# Development and evaluation of a machine learning-based in-hospital COVID-19 disease outcome predictor (CODOP): A multicontinental retrospective study

Riku Klén[1], Disha Purohit[2], Ricardo Gómez-Huelgas[3], José Manuel Casas-Rojo[4], Juan Miguel Antón-Santos[4], Jesús Millán Núñez-Cortés[5], Carlos Lumbreras[6], José Manuel Ramos-Rincón[7], Noelia García Barrio[8], Miguel Pedrera-Jiménez[8], Antonio Lalueza Blanco[6], María Dolores Martin-Escalante[9], Francisco Rivas-Ruiz[10], Maria Ángeles Onieva-García[11], Pablo Young[12], Juan Ignacio Ramirez[12], Estela Edith Titto Omonte[13], Rosmery Gross Artega[14], Magdy Teresa Canales Beltrán[15], Pascual Ruben Valdez[16], Florencia Pugliese[16], Rosa Castagna[16], Ivan A Huespe[17], Bruno Boietti[17], Javier A Pollan[17], Nico Funke[18], Benjamin Leiding[19], David Gómez-Varela[18,20]*

[1]Turku PET Centre, University of Turku and Turku University Hospital, Turku, Finland; [2]Max Planck Institute of Experimental Medicine, Göttingen, Germany; [3]Internal Medicine Department, Regional University Hospital of Málaga, Biomedical Research Institute of Málaga (IBIMA), University of Málaga (UMA), Málaga, Spain; [4]Internal Medicine Department, Infanta Cristina University Hospital, Madrid, Spain; [5]Internal Medicine Department, Gregorio Marañón University Hospital, Madrid, Spain; [6]Internal Medicine Department, 12 de Octubre University Hospital, Madrid, Spain; [7]Internal Medicine Department, General University Hospital of Alicante, Alicante Institute for 22 Health and Biomedical Research (ISABIAL), Alicante, Spain; [8]Data Science Unit, Research Institute Hospital 12 de Octubre, Madrid, Spain; [9]Internal Medicine Department, Hospital Costa del Sol, Marbella, Spain; [10]Hospital Costa del Sol. Research Unit, Marbella, Spain; [11]Preventive Medicine Department, Hospital Costa del Sol, Marbella, Spain; [12]Hospital Británico of Buenos Aires, Buenos Aires, Argentina; [13]Internal Medicine Service, Hospital Santa Cruz - Caja Petrolera de Salud, Santa Cruz, Bolivia; [14]Epidemiology Unit, Hospital of San Juan de Dios, Santa Cruz, Bolivia; [15]Instituto Hondureno of social security, Hospital Honduras Medical Centre, Tegucigalpa, Honduras; [16]Hospital Velez Sarsfield, Buenos Aires, Argentina; [17]Hospital Italiano de Buenos Aires, Buenos Aires, Argentina; [18]Max Planck Institute for Experimental Medicine, Göttingen, Germany; [19]Institute for Software and Systems Engineering at TU Clausthal, Clausthal, Germany; [20]Systems Biology of Pain, Division of Pharmacology & Toxicology, Department of Pharmaceutical Sciences, University of Vienna, Vienna, Austria

*For correspondence:
david.gomez.varela@univie.ac.at

**Competing interest:** The authors declare that no competing interests exist.

**Abstract** New SARS-CoV-2 variants, breakthrough infections, waning immunity, and sub-optimal vaccination rates account for surges of hospitalizations and deaths. There is an urgent need for clinically valuable and generalizable triage tools assisting the allocation of hospital resources, particularly in resource-limited countries. We developed and validate CODOP, a machine learning-based tool for predicting the clinical outcome of hospitalized COVID-19 patients. CODOP was trained, tested and

validated with six cohorts encompassing 29223 COVID-19 patients from more than 150 hospitals in Spain, the USA and Latin America during 2020–22. CODOP uses 12 clinical parameters commonly measured at hospital admission for reaching high discriminative ability up to 9 days before clinical resolution (AUROC: 0·90–0·96), it is well calibrated, and it enables an effective dynamic risk stratification during hospitalization. Furthermore, CODOP maintains its predictive ability independently of the virus variant and the vaccination status. To reckon with the fluctuating pressure levels in hospitals during the pandemic, we offer two online CODOP calculators, suited for undertriage or overtriage scenarios, validated with a cohort of patients from 42 hospitals in three Latin American countries (78–100% sensitivity and 89–97% specificity). The performance of CODOP in heterogeneous and geographically disperse patient cohorts and the easiness of use strongly suggest its clinical utility, particularly in resource-limited countries.

## Editor's evaluation

This submission is dealing with the unmet need to generate a machine learning approach for the early and accurate estimation of the outcome of patients admitted to hospital with COVID-19. The presented data generate confidence on the validity since they have been developed based on a vast number of patients and the data are validated in cohorts from different geographical regions.

## Introduction

Since the first reported case in Wuhan at the end of 2019, COVID-19 has exerted extreme pressure on hospitals throughout the globe. The World Health Organization (WHO) estimated the pandemic as the direct cause of more than six million deaths. Despite the decreased number of hospitalizations and deaths among vaccinated people, warning signs forecast a scenario with health systems under severe strains leading to a bigger number of COVID-19 related deaths. The appearance of viral variants that are more contagious and that carry a higher risk of hospitalization, (*Twohig et al., 2022*) the waning of the immune protection, the significant amount of infections in vaccinated individuals (breakthrough infections) (*Shen, 2022*) together with their ability to transmit the virus, and the slow and unequal rollout of vaccines worldwide, support recent models showing that a vaccine-alone exit strategy will likely not be sufficient to contain further outbreaks and their consequences (*Moore et al., 2021*) At the time of submission of this study, many countries are reaching record-high numbers of infections, hospitalizations and deaths. This new pandemic wave depicts a worrisome prospect for resource-limited countries with similar or lower vaccination rates and with fewer clinical tools.

Prediction models that estimate the risk of death in hospitalized COVID-19 patients could be valuable both to clinicians and patients by assisting medical staff to stratify treatment strategies and by planning for the appropriate allocation of limited resources. Thus, numerous models have been developed to assist in triage decisions of hospitalized COVID-19 patients. However, independent evaluations have pointed out their lack of generalizability and their limited clinical use (*Wynants et al., 2020*; *El-Solh et al., 2020*) due to causes belonging to the 'dataset shift' problem (*Subbaswamy and Saria, 2020*). Moreover, the heterogeneity of the host-pathogen interaction (what results in more than 60 disease subtypes of COVID-19 *DeMerle et al., 2021*) together with the fast evolution of the pandemic makes COVID-19 outcome prediction a challenging endeavour, especially if a profound evaluation using patient cohorts from geographically distinct regions is not performed. Finally, the effectiveness of these predictive models in patients with diverse immune protection (due to natural infection or vaccination) and patients infected by different Variants Of Concern (VOC) is unknown.

To address this need, we used the largest and the most geographically extended patient dataset to date for developing and extensively validating a simple yet clinically useful machine learning-based online model for doctors to predict mortality in COVID-19 patients at any time during hospitalization. To assist the real clinical needs during different pandemic scenarios we offer two predictor subtypes suited for undertriage and overtriage situations (https://gomezvarelalab.em.mpg.de/codop/).

The collective effort presented here unveils the power of machine learning for helping clinicians and patients in this pandemic. Based on its easiness to use and its generalizability among geographically very distinct patient cohorts, we aim for CODOP to become a useful triage tool, particularly in resource-limited countries.

**eLife digest** While COVID-19 vaccines have saved millions of lives, new variants, waxing immunity, unequal rollout and relaxation of mitigation strategies mean that the pandemic will keep on sending shockwaves across healthcare systems. In this context, it is crucial to equip clinicians with tools to triage COVID-19 patients and forecast who will experience the worst forms of the disease. Prediction models based on artificial intelligence could help in this effort, but the task is not straightforward.

Indeed, the pandemic is defined by ever-changing factors which artificial intelligence needs to cope with. To be useful in the clinic, a prediction model should make accurate prediction regardless of hospital location, viral variants or vaccination and immunity statuses. It should also be able to adapt its output to the level of resources available in a hospital at any given time. Finally, these tools need to seamlessly integrate into clinical workflows to not burden clinicians.

In response, Klén et al. built CODOP, a freely available prediction algorithm that calculates the death risk of patients hospitalized with COVID-19 (https://gomezvarelalab.em.mpg.de/codop/). This model was designed based on biochemical data from routine blood analyses of COVID-19 patients. Crucially, the dataset included 30,000 individuals from 150 hospitals in Spain, the United States, Honduras, Bolivia and Argentina, sampled between March 2020 and February 2022 and carrying most of the main COVID-19 variants (from the original Wuhan version to Omicron). CODOP can predict the death or survival of hospitalized patients with high accuracy up to nine days before the clinical outcome occurs. These forecasting abilities are preserved independently of vaccination status or viral variant.

The next step is to tailor the model to the current pandemic situation, which features increasing numbers of infected people as well as accumulating immune protection in the overall population. Further development will refine CODOP so that the algorithm can detect who will need hospitalisation in the next 24 hours, and who will need admission in intensive care in the next two days. Equipping primary care settings and hospitals with these tools will help to restore previous standards of health care during the upcoming waves of infections, particularly in countries with limited resources.

## Materials and methods
### Patient cohorts
The training and two test cohorts (Test 1 and Test 2) of this study are based on the SEMI (Sociedad Espanola de Medicina Interna) COVID-19 Registry (*Casas-Rojo et al., 2020*) It is an ongoing multicentre nationwide cohort of consecutive patients hospitalized for COVID-19 across different Spanish regions (109 hospitals). Eligibility criteria were age ≥18 years, confirmed diagnosis of COVID-19, defined as a positive result on real-time reverse-transcription-polymerase-chain-reaction (RT-PCR) for the presence of SARS-CoV-2 in nasopharyngeal swab specimens or sputum samples, first hospital admission for COVID-19, and hospital discharge or in-hospital death (*Casas-Rojo et al., 2020*).

An additional patient cohort (named Test 4 and composed of 2508 patients hospitalized in the 12 de Octubre and the Costa del Sol Spanish hospitals), was used for testing the influence of vaccination and the Delta and Omicron virus variants on the discriminative ability of CODOP for predicting in-hospital death, the need for mechanical ventilation and admission to the Intensive Care Unit (ICU).

Personal data are processed in strict compliance with Spanish Law 14/2007, of July 3, on Biomedical Research, Regulation (EU) 2016/679 of the European Parliament and of the Council of 27 April 2016 on the protection of natural persons with regard to the processing of personal data and on the free movement of such data, repealing Directive95/46/EC (General Data Protection Regulation), and Spanish Organic Law 3/2018, of December 5, on the Protection of Personal Data and the Guarantee of Digital Rights. The SEMI-COVID-19 Registry and the COVID registries of 12 de Octubre and the Costa del Sol hospitals has been approved by the Provincial Research Ethics Committee of Malaga (Spain; C.I.F. number: 0–9150013-B).

In accordance with applicable regulations, the Spanish Agency of Medicines and Medical Products (AEMPS, for its initials in Spanish) has ruled that due to its nature, the study only required the approval of the Ethics Committee and not the Autonomous Community, as in other studies.

**Table 1.** Features used during CODOP development with the training cohort, the values used for imputation, and the percentage of missing values.

Numerical variables are reported by median (Md) and interquartile range (IQR).

| Variable | Imputed value | Md (IQR) | Missing % |
|---|---|---|---|
| Age (years) | 66·67,911 | 68 (56–79) | 0·0 |
| Sex (male, female) | none | 6 775 females and 9 127 males | 0·0 |
| Hemoglobin (g/dL) | 13·33,201 | 13 (12–15) | 1·7 |
| Platelet Count (x 10⁶ /L) | 250 097·7 | 223,000 (164 000-311 000) | 1·8 |
| Eosinophils (x 10⁶ /L) | 63·81,817 | 10 (0–100) | 3·0 |
| Lymphocytes (x 10⁶ /L) | 1 243·575 | 1,000 (700-1 420) | 1·9 |
| Neutrophils (x 10⁶ /L) | 5 525·894 | 4 490 (3 090-6 800) | 2·2 |
| Monocytes (x 10⁶ /L) | 535·8,804 | 470 (300–660) | 2·7 |
| C-Reactive Protein (mg/L) | 74·48,964 | 41 (12–108) | 4·6 |
| Creatinine (mg/dL) | 1·156,574 | 1 (1–1) | 2·0 |
| Lactate Dehydrogenase (U/L) | 363·9,083 | 306 (234–424) | 13·0 |
| Aspartate aminotransferase (U/L) | 49·27,098 | 35 (24–53) | 18·4 |
| Alanine aminotransferase (U/L) | 48·99,699 | 32 (20–54) | 7·4 |
| Total bilirrubin (mg/dL) | 0·6429202 | 1 (0–1) | 26·5 |
| Serum Sodium (mmol/L) | 138·4,268 | 138 (136–141) | 2·6 |
| Serum Potassium (mmol/L) | 4·178,441 | 4 (4–4) | 3·7 |
| Glucose (mg/dL) | 124·2,852 | 108 (92–135) | 5·2 |
| Prothrombin time (s) | 19·99,798 | 13 (12–14) | 35·8 |
| Fibrinogen (mg/dL) | 608·0043 | 601 (497–713) | 37·0 |
| Dimer (ng/mL) | 2 122·158 | 672 (370–1 320) | 21·7 |

The test cohort from New York (External Test 3) is based on the study from *Del Valle et al., 2020* consisting of 2 021 COVID-19 patients hospitalized in the Mount Sinai Health System in New York City.

The cohorts used in the online validation of the two online CODOP subtypes were provided by a group of Argentinian hospitals composed by the Argentinian COVID-19 Network (4690 patients from 37 Argentinian hospitals), Hospital Vélez Sarsfield (100 patients, Buenos Aires, Argentina), and Hospital Británico de Buenos Aires (150 patients, Buenos Aires, Argentina), the Honduras Medical Centre (45 patients, Tegucigalpa, Honduras), the Hospital Santa Cruz Caja Petrolera de Salud (30 patients, Santa Cruz de la Sierra, Bolivia), and the Hospital San Juan de Dios (93 patients, Santa Cruz, Bolivia). The Argentinian COVID-19 Network was also used for predicting ICU admission and the need for mechanical ventilation.

Process of personal data are in strict compliance with National Laws of personal data and in accordance with the principles of the Declaration of Helsinki. The release of anonymized clinical data used in this study has been reviewed by the institutional ethical review boards for each institution participating in this study (approval numbers: 1575, 5562, and 5606 for the Argentinian datasets, 143-CB-HE for Honduras Medical Centre). For the Hospital Santa Cruz Caja Petrolera de Salud and Hospital San Juan de Dios, please contact corresponding authors for additional details regarding the IRB approval documents. Informed consent to publish their de-identified clinical data for academic purposes was obtained from all the patients. When it was not possible to obtain informed consent in writing due to biosafety concerns or if the patient had already been discharged, informed consent was requested verbally and noted on the medical record.

## Predictors and outcomes

We included patient characteristics and blood test values (see *Table 1*) that were present in all training and test cohorts, measured at different times during hospitalization, as potential predictors. We

limited our potential predictors to variables that had less than 40% missing values. The percentage of missing values is listed in *Table 1*. Most of the variables have less than 5% of missing values. Missing values were imputed in all datasets using the mean value of original variables in the training cohort. We trained a binary classification model in which the outcome is patient mortality: 1, if the patient was deceased, or 0, if discharged.

For each cohort, the subjects were divided into two groups based on their survival status. The normality of each numerical variable in the groups was tested with the Shapiro-Wilk normality test. None of the variables was normally distributed. For each variable, statistical difference was tested between the two groups with the Wilcoxon rank-sum test for numerical variables and with the chi-squared test for categorical variables. The obtained p-values were adjusted for multiple testing by Benjamini-Hochberg Procedure.

Models for both the need of mechanical ventilation and admission to the ICU were constructed in a similar fashion.

## CODOP development

CODOP was built using modified stable iterative variable selection (SIVS) (*Mahmoudian et al., 2021*) and linear regression with least absolute shrinkage and selection operator (lasso) regularisation (*Friedman et al., 2010*). In model building only the training cohort was used and models were built using 10-fold cross-validation. In the feature selection stage of SIVS, 100 models were built and for each model selected variables were recorded. For reducing the number of features to as few as possible (therefore, increasing the easiness of use of CODOP), we tuned the weighting function in SIVS (called variable importance scoring) so that only features occurring in all of the 100 models were selected for the final model building stage. This method has shown to be very efficient, especially when the ratio of positive and negative outcomes is imbalanced (*Klén et al., 2019*). Lasso models were built in *R Development Core Team, 2010* (version 3.6.0) package glmnet (*Friedman et al., 2010*) (version 4.1–1). All predictions were done blinded to the final clinical outcome. For converting numeric prediction into binary prediction, Youden's J statistic was used (*Youden, 2006*). For building the two online CODOP subtypes, we used alternative thresholds, which were selected to be the largest threshold value in the training cohort with a sensitivity of 95% for CODOP-Ovt and specificity of 95% for CODOP-Unt. Calibration plots were created with R package caret (*Kuhn, 2020*) (version 6.0–86). Survival analysis was performed using univariable Cox proportional hazards regression model (*Cox, 1972*). Survival analysis and Kaplan-Meyer plots were produced with R packages survival (*Therneau and Lumley, 2020*) (R package version 3.2–11) and survminer (*Alboukadel et al., 2018*) (R package version 0.4.9). For horizon analyses, the data were considered separately for survival time of one to nine days.

The final model can be found in the Klén etal_*Supplementary file 1* and it is freely accessible in the following Github addresses: https://github.com/TUC-Circular-Economy-Department/COvid-19-Disease-Outcome-Predictor#uir https://github.com/TUC-Circular-Economy-Department/COvid-19-Disease-Outcome-Predictor#documentation.

## Benchmarking

To evaluate the performance of CODOP, we used three benchmark methods: COPE (*van Klaveren et al., 2021*), model by *Zhang et al., 2020*, and a univariable model. COPE model is a linear regression model, which uses variables age, respiratory rate, C-reactive protein, lactic dehydrogenase, albumin, and urea. Zhang et al. model is a logistic regression model, which uses variables age, sex, neutrophil count, lymphocyte, platelet, C-reactive protein, and creatinine. From the different models described in Zhang et al., model DL for prediction of death (Table S2 of Zhang et al.) was used for benchmarking purposes. Univariable analysis was performed in the training dataset for all variables. The best univariable model was selected based on the average ranking of AUROC, accuracy, sensitivity and specificity. Different models were evaluated using four evaluation metrics: area under receiver operating curves (AUROC), accuracy, sensitivity, and specificity. The metrics were calculated using R packages pROC (*Robin et al., 2011*) (version 1.17.0.1) and caret (*Kuhn, 2020* R package version 6.0–86).

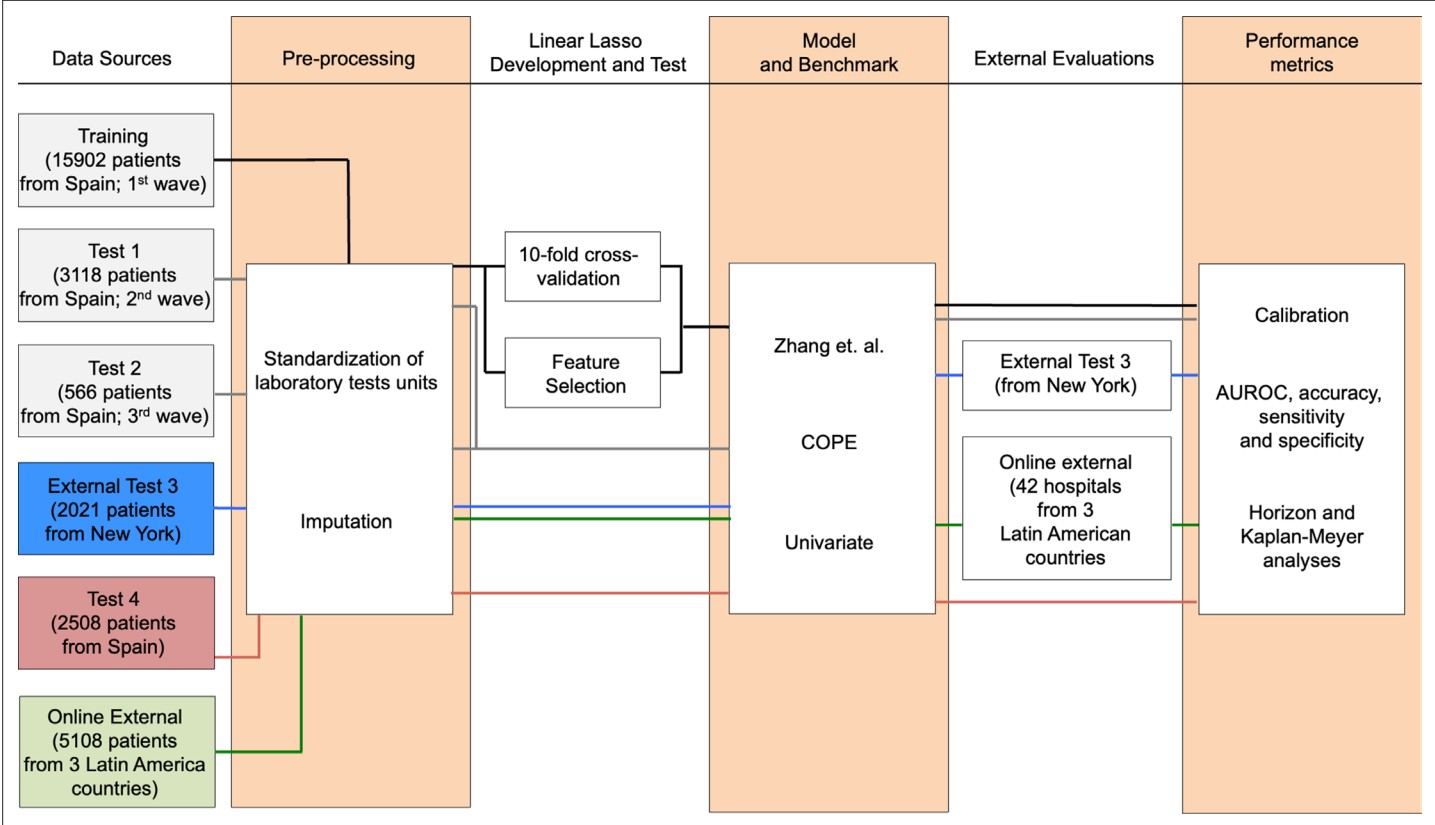

**Figure 1.** Flowchart depicting the different patient cohorts used in this study and the steps followed during the development, test, and independent evaluation of CODOP.

## Online evaluation

Forty-two different Latin American hospitals provided the values for the 12 features used by CODOP that were measured in patients at two different time points between March 7th 2020 and October 16th 2021: during the time of hospitalization, and the worst values measured during hospitalization. The former datasets were used for calculating AUROC, calibration curves, and confusion matrices. Both times points were used for performing horizon analysis and risk-stratification. All predictions were done blinded to the final clinical outcome.

## Role of the funding source

The Max Planck Society support the payment of the article processing fees. No other funding supported the study. The funders of the had no role in study design, data collection, data analysis, interpretation of data, writing of the report, or in the decision to submit the paper for publication.

## Results

### CODOP development, performance, and benchmark

We developed CODOP following a multistep process (*Figure 1*) using a training dataset with measurements of 20 features (18 blood biochemical parameters plus Age and Sex; *Table 1*) routinely measured during admission on 15902 COVID-19 patients hospitalized in 109 Spanish healthcare centres during the first COVID-19 wave that occurred in Spain between February 5th and July 6th 2020 (SEMI-COVID-19 Network database *Casas-Rojo et al., 2020*).

As a first step, data pre-processing included standardization of the laboratory tests units and imputation of the missing test values, which is characteristic of real-world clinical practice (*Table 1*). Using linear Lasso, 10-fold cross-validation and SIVS, we obtained a final CODOP model using 11 blood biochemical parameters plus Age (*Supplementary file 1* and *Figure 2—figure supplement*

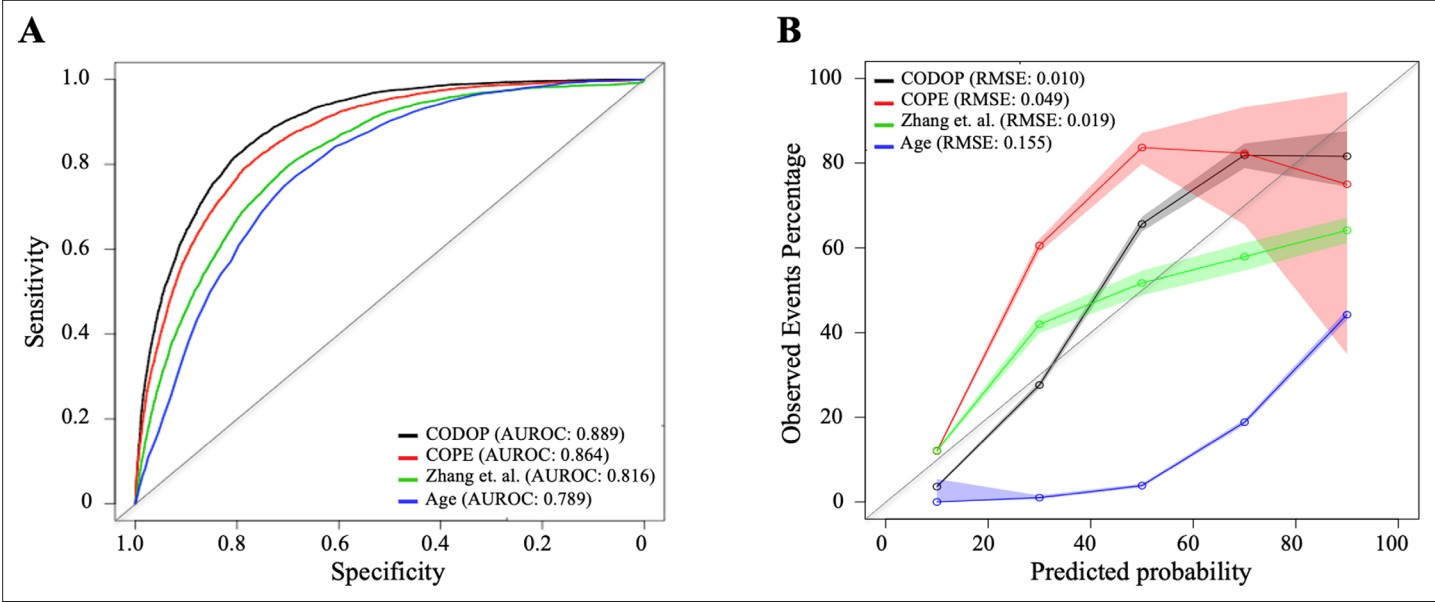

**Figure 2.** Discriminatory ability (using area under the receiver operating curves or AUROC; **A**) and calibration curves (**B**) for CODOP, COPE, Zhang et al., and Age in the training dataset.

The online version of this article includes the following source data and figure supplement(s) for figure 2:

**Source data 1.** Prediction values for CODOP, COPE, Zhang et.

**Figure supplement 1.** Optimisation of the final COPOD model by selecting predictors using the least absolute shrinkage and selection operator (LASSO) method.

**Figure supplement 1—source data 1.** Mean squared error and the Penalty parameter ($\lambda$).

**Figure supplement 2.** Discriminatory ability (using area under receiver operating curves or AUROC) (A) and calibration curves (**B**) for CODOP, COPE, Zhang et al., and Age in the test datasets.

**Figure supplement 2—source data 1.** Prediction values for CODOP, COPE, Zhang et.

**Figure supplement 3.** Discriminatory ability of CODOP (using area under receiver operating curves or AUROC) taking into account the Delta and Omicron VOCs (**A**) and the vaccination status of the patients (**B**) in the Test 4 dataset.

**Figure supplement 3—source data 1.** Prediction values for CODOP in vaccinated individuals and in patients infected with the Delta or Omicron virus variants.

*1*). Detailed analysis indicated elevated values of Age, neutrophils, C-reactive protein, creatinine, lactate dehydrogenase, serum sodium, serum potassium, glucose and D-dimer, and reduced values of platelets, eosinophils and monocytes were positively correlated with in-hospital death, respectively (*Supplementary file 1*).

Next, we benchmarked the performance of CODOP, using the same training dataset, against the predictor developed by *Zhang et al., 2020*, against the predictor COPE (*van Klaveren et al., 2021*), and against Age (as the univariable feature with more predictive power; *Supplementary file 1*). The two prognostic models were selected based on the availability of the model's details and their use of blood-based features. CODOP showed a superior discriminative ability in predicting in-hospital mortality (area under the receiver operating curves or AUROC: 0·889, 95% CI 0·885–0·894; *Figure 2A*) reaching 0·84% and 0·78% sensitivity and specificity, respectively (*Supplementary file 1*). In addition, CODOP has better calibration for all the different risk groups as reflected by a lower RMSE value (*Figure 2B* and *Supplementary file 1*). A detailed inspection of the calibration curves shows that the predictor published by Zhang et al. underestimated the probability of death for low-risk patients and overestimates the probability of death for high-risk patients. On the other side, while COPE underestimates the probability of death for all risk groups, Age showed a clear overestimation (*Figure 2B*).

## Influence of the geographical location, the vaccination status and the type of VOCs in the discriminative ability of CODOP

The size, demographic diversity (in terms of age, gender, ethnicity and comorbidities; see *Table 1* of *Casas-Rojo et al., 2020*), and geographical spread of the training dataset, suggest the generalizability of the predictions made by CODOP. However, the rapid evolution of the pandemic challenges any prediction model that relies on past datasets. We investigated the discriminative ability and calibration of CODOP in geographical diverse patient cohorts having different vaccination statuses and infected with different VOCs.

On the one side, we used two time-sliced cohorts with COVID-19 patients hospitalized during consecutive COVID-19 waves that occurred in Spain between July 7th and December 6th 2020 (Test 1; 3118 patients), between December 7th 2020 and March 31st 2021 (Test 2; 566 patients). These cohorts are composed of non-vaccinated patients infected with the original Wuhan and the Alpha virus variants. ROC and calibration curves show that the performance metrics are preserved in these two cohorts (*Figure 2—figure supplement 2*, *Supplementary file 1*). The generalizability of CODOP was also demonstrated on a separate test cohort (External Test (3) consisting of 2021 COVID-19 patients hospitalized in the Mount Sinai Health System in New York City between March 21st and April 28th, 2020 *Del Valle et al., 2020*; *Figure 2—figure supplement 2*, *Supplementary file 1*).

To investigate the influence of the Delta and Omicron VOCs, we analysed a dataset collected in Spain between April 1st 2021 and February 27th 2022 (Test 4; 2508 patients). Our data demonstrate that the performance of CODOP is preserved when these two types of VOC were dominant (*Figure 2—figure supplement 2*, *Supplementary file 1*). A detailed analysis shows that CODOP has the same discriminative ability in patients infected by Delta or by Omicron (*Figure 2—figure supplement 3*). The lack of correlation between the percentage of deaths and the predictive ability of CODOP among the four test cohorts (*Supplementary file 1*), rules out an artefactual influence due to the time-sliced nature of these cohorts. Finally, CODOP overperformed both the benchmarked predictors and Age in the test cohorts (*Figure 2—figure supplement 2*, *Supplementary file 1*).

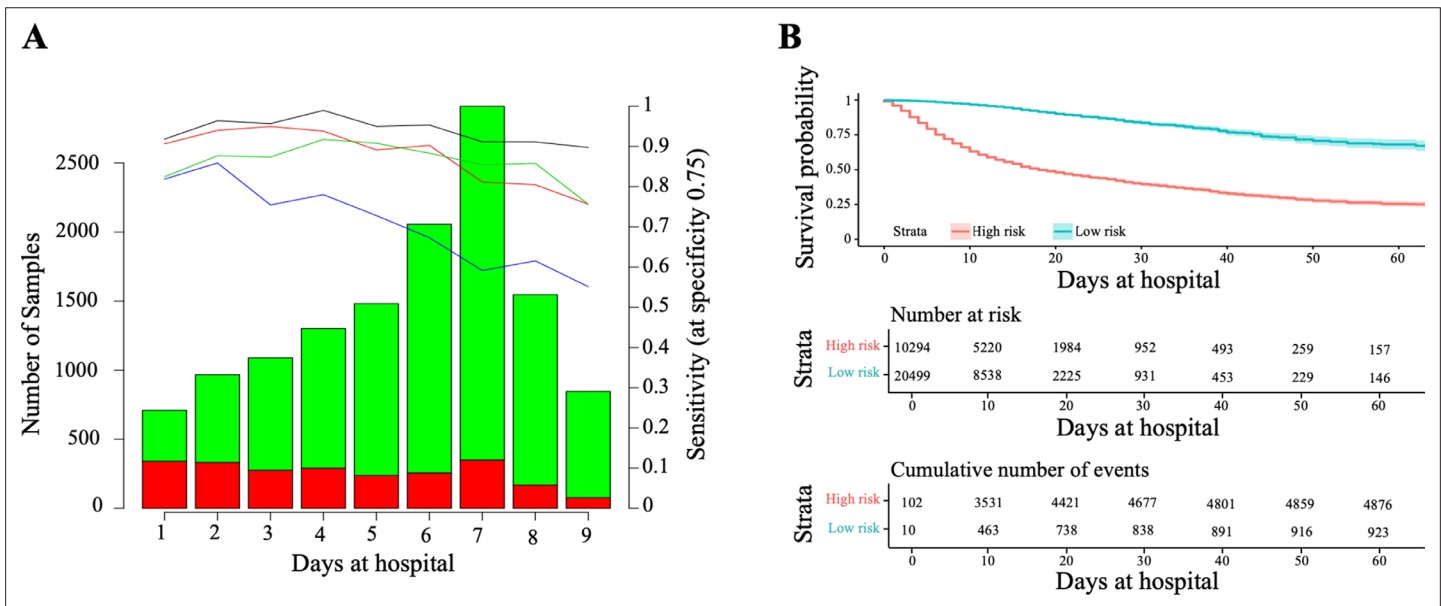

**Figure 3.** Horizon analysis (**A**) and survival analysis (**B**) in the training dataset. In the horizon plot, x-axis represents the number of days at the hospital before clinical resolution, the bar plot is for the number of samples (the green colour is for survival and red for death), and lines are for sensitivity when the specificity was fixed at 75% in the training cohort (the black line is CODOP, the red line is COPE, the green line is Zhang et al., and the blue line is Age). In the survival analysis, the risk scores refer to the probability provided by CODOP.

The online version of this article includes the following source data and figure supplement(s) for figure 3:

**Source data 1.** Prediction values for CODOP, COPE, Zhang et.

**Figure supplement 1.** Survival analysis in the test datasets.

**Figure supplement 1—source data 1.** Prediction values for CODOP, COPE, Zhang et.

To evaluate the discriminative ability of CODOP in the vaccinated population we analyzed data from hospitalized patients with two or three vaccination doses, belonging to the Test 4 cohort. Our data show that vaccination status has little influence on the ability of CODOP for predicting the risk of in-hospital death (*Figure 2—figure supplement 3* and *Supplementary file 1*).

Altogether, our results show the stability of CODOP during the fluctuating scenario of the COVID-19 pandemic (the appearance of different VOCs, the different immune protection among the population, the use of more tailored clinical interventions), suggesting that CODOP captures key biomarkers involved in the physiological deterioration of COVID-19 hospitalized patients.

## Estimation of fixed prediction horizons and dynamic risk-stratification

Many patients of the different cohorts had multiple blood samples taken during their hospitalization. This offers a possibility for investigating the time window, before clinical resolution, at which CODOP can predict the death of hospitalized patients with high sensitivity.

For that, we compared the performance of CODOP at a fixed time before the clinical resolution using the training cohort. On average, CODOP predicted the outcome of all patients nine days in advance with an average sensitivity (at a fixed specificity of 75%) and AUROC values higher than 90% (*Figure 3A* and *Supplementary file 1*, respectively). In comparison to the other benchmarked predictors, CODOP maintained a stable sensitivity along the nine days horizon time significantly outperforming (*Figure 3A*; p < 0.01, paired two-sided T-test).

Next, we demonstrated that CODOP enables a continuous stratification of patients into a high-risk group over the course of the hospitalization, as patients with a higher risk score (which refers to the probability provided by CODOP), who were more likely to die over time (*Figure 3B*). We obtained similar stratification results when using other test cohorts (*Figure 3—figure supplement 1*). Hence, CODOP represents an early and dynamic warning tool in the clinical status of COVID-19 patients.

## Multinational evaluation of an online CODOP predictor

During the COVID-19 pandemic, the availability of resources in hospitals around the world experiences significant fluctuations following successive infection waves. Thus, a clinically useful prediction tool needs to reckon with these dynamic scenarios for effectively assisting undertriage and overtriage decisions.

We developed and validated two subtypes of our predictor, CODOP-Ovt (from overtriage) and CODOP-Unt (from undertriage), intending to optimize the triage of patients at high risk of death upon arrival to the hospital and after their first blood analysis. CODOP-Ovt maximizes the negative predictive value or the detection of high-risk patients (high sensitivity) and it is meant for scenarios where overtriage is possible because hospital resources are not the main limitation. On the other side, CODOP-Unt maximizes the positive predictive value by trying to avoid the inclusion of false high-risk patients (high specificity) and it might be preferred in pandemic conditions when hospital resources are limited and undertriage needs to be considered.

Using the initial training cohort, CODOP-Ovt identified >95% of the patients that finally died in hospital nine days before clinical resolution (*Figure 4—figure supplement 1*). As expected, this increase in sensitivity is concomitant with reduced specificity (60%–70%; *Figure 4—figure supplement 1*). Notably, these metrics are within the range of recommended under- and overtriage levels ranging from 5% to 10% and 25% to 50%, respectively (*van Rein et al., 2019*) The opposite results were obtained with CODOP-Unt, where more than 95% of the patients that survived were correctly predicted as low-risk (*Figure 4—figure supplement 1*) while 40%–50% of the patients that died in hospital were not detected in advance (*Figure 4—figure supplement 1*). Confusion matrixes show similar overall performance for both CODOP subtypes in all test cohorts (*Supplementary file 1*).

Following this, we constructed and evaluated an easy-to-use web-based application (https://gomezvarelalab.em.mpg.de/codop/) that offers the possibility to choose between CODOP-Ovt and CODOP-Unt. The web application includes a detailed description of the CODOP project and instructions on how to use the prediction tool. The web application has been tested using different devices, web browsers and operative systems (*Supplementary file 1*). In all cases, predictions were calculated in less than 2 s for datasets up to 2000 patients (data not shown). Further, the Data Protection Office of the Max Planck Society assisted in assuring the legal fit of the web application to the General Data Protection Regulation (GDPR).

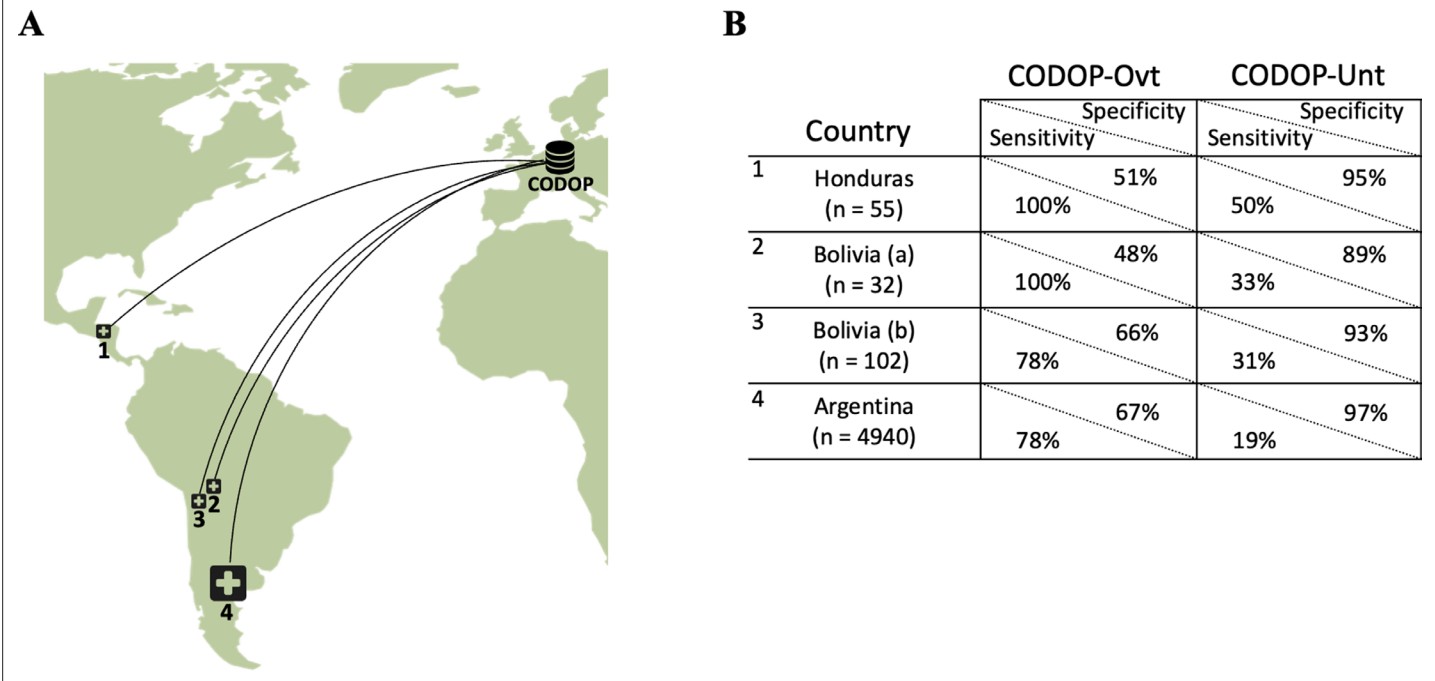

**Figure 4.** The geographical location of the external cohorts from 42 different Latin American hospitals used during the online evaluations (**A**) and performance of web calculators CODOP-Ovt and CODOP-Unt in these external cohorts number of patients from each institution are indicated in parenthesis; (**B**).

The online version of this article includes the following source data and figure supplement(s) for figure 4:

**Source data 1.** Prediction values for CODOP in the Latin American cohort.

**Figure supplement 1.** Horizon analysis in the training dataset for sensitivity (**A**) and specificity (**B**).

**Figure supplement 1—source data 1.** Prediction values for CODOP-Ovt and CODOP-Unt for the horizon analysis in the training dataset.

To make a stringent external evaluation of this application with datasets collected from very different patient cohorts, we established a multinational collaboration with 42 hospitals from three Latin American countries (*Figure 4A*), which at the time of this evaluation were under a new surge of COVID-19 infections and admissions coinciding with the beginning of the Autumn-Winter season in the Southern Hemisphere. All these hospitals provided the values for the 12 features used by CODOP and measured in patients at the time of hospitalization between March 7th 2020 to October 16th 2021. Following, these data were uploaded to the two CODOP online subtypes and we obtained the mortality predictions that were compared to the real patient outcome (for which the online predictor was blinded).

Importantly, AUROC values for CODOP-Ovt and CODOP-Unt demonstrate the generalizability of the predictor (*Supplementary file 1*). A detailed analysis of the results indicates that if these were a prospective study, CODOP-Ovt would have identified the majority of the patients that finally died during hospitalization albeit wrongly classifying them as high-risk a significant number of patients that finally survived (78–100% sensitivity and 48–67% specificity, respectively; *Figure 4B* and *Supplementary file 1*). On the other side, the use of CODOP-Unt would have correctly triaged the vast majority of the survivors despite missing a significant number of patients that finally died (89–97% specificity and 19%–50% sensitivity, respectively; *Figure 4B* and *Supplementary file 1*). These results strongly suggest that the online version of CODOP could represent a useful clinical tool in the triage decision protocols.

## Discussion

The differential access to COVID-19 vaccines, the emergence of new viral variants of concern, the waning of the immune protection, and the relaxation of mitigation measurements anticipate a longer period of health systems under pressure due to an increasing number of COVID-19 patients,

particularly in resource-limited countries. A conflagration-like scenario will likely be the final set of the pandemic for many nations (*Kofman et al., 2021*) As a result of an altruistic multicontinental effort, we developed and evaluated CODOP, a machine-learning-based online tool able to assist in triage decisions in hospitalized COVID-19 patients. CODOP uses 12 clinical parameters easy to collect in most hospitals. Its predictive performance among cohorts of patients with different geographical locations, vaccination statuses and infected by diverse VOCs, strongly suggests its generalizability and supports its potential for improving patient care during this pandemic.

CODOP satisfies the Transparent Reporting of a multivariable prediction model for Individual Prognosis Or Diagnosis principles (*Collins et al., 2015*) (TRIPOD; *Supplementary file 1*), follows the recently proposed MINimum Information for Medical AI Reporting (*Hernandez-Boussard et al., 2020*) (MINIMAR; *Supplementary file 1*), and it has been successfully checked for the risk of bias and applicability using the Prediction model study Risk of Bias Assessment Tool (*Moons et al., 2019*) (PROBAST; *Supplementary file 1*).

The use of such an early warning system as CODOP could potentially represent an important help in clinical decision-making including the prioritization of care and resource allocation. The novelty of the COVID-19 disease and its toll on the health systems has led to dozens of triage policies, many of them based on some form of Sequential Organ Failure Assessment (SOFA) scores. (*Raschke et al., 2021*) In addition, several machine learning-based prediction tools have been developed during this pandemic. However, independent validation studies have dismissed the clinical utility of all these models (*Wynants et al., 2020*; *El-Solh et al., 2020*) and have indicated common pitfalls to be avoided such as small sample size, use of variables not easily measurable in most hospitals, and lack of external evaluation datasets gathered in geographically different cohorts, etc. To avoid this 'dataset shift' problem and aim to increase the generalization of CODOP, we set to satisfy the so-called stability property (*Subbaswamy and Saria, 2020*) For this we used an initial training and test cohorts encompassing 24,345 patients from more than 110 hospitals spread over Spain and the USA and gathered during three pandemic waves. Both the size, heterogeneity of the patient population (in terms of age range, ethnicity, comorbidities, etc.), and the myriad of clinical and analytical procedures performed during the pandemic, ensures a significant number of perturbations (shifts) in how the data were generated. This strategy seems to be supported by the stable performance of our predictor on the external online evaluation performed with a patient cohort gathered in 42 hospitals in three Latin American countries. Importantly, we demonstrate that the discriminative ability of CODOP is not hampered by the different immune protection status (either by infection or by vaccination) or by the type of VOC, which suggests that the inflammatory process underlying the clinical manifestations is similar in most of the hospitalized patients independently of their vaccination status or the type of VOC.

In addition to the characteristics of our cohorts, we hypothesized that the higher performance achieved by CODOP when compared to published mortality risk scores is due to the use of a group of biochemical parameters representing the main biological pathways involved in the pathogenesis of SARS-CoV-2. A very common clinical manifestation in critical COVID-19 patients is composed of a deregulated immune response and a robust inflammatory reaction (known as 'hypercytokinemia' or 'cytokine storm'), which ultimately leads to tissue injury (*Chen and Quach, 2021*) Recent reports show a downregulated type-I interferon response leading to an increase of neutrophils in severe COVID-19 patients (*Zhang et al., 2021*). These findings go in line with our data showing alterations in several myeloid cells (eosinophils, monocytes) including an upregulation in the number of neutrophils (*Supplementary file 1b*). Myeloid cells are crucial for mounting a successful immune response against viruses and for the existence of hypercytokinemia (*Bordon et al., 2013*) The increased level of CRP and LDH in our dataset and their predictive value could represent easy-to-measure hallmarks of the exacerbated inflammatory response associated with a high risk of COVID-19-related death. These and other model features linked to thromboembolic complications (i.e. D-dimer and Platelets) and organ failure (i.e. Creatinine), could represent a warning signature easy to evaluate at early stages of the infection, even before failure in major functions can be monitored.

Several unique biomarkers have been suggested as surrogates for guiding clinical decisions in COVID-19 patients. As example, C-reactive Protein (CRP) is useful to recognize patients with an increased risk of mortality during hospitalization (*Stringer et al., 2021*). However, the 12 parameter CODOP model has higher predictive performance than the CRP when analyzed as univariate

(*Supplementary file 1*). For other biomarkers (e.g. Interleukin 6 *Coomes and Haghbayan, 2020* or suPAR *Altintas et al., 2021*), the elevated price of the analysis precludes its widely use in Emergency Departments, particularly in low-income countries. The advantage of a multi-parameter predictor like CODOP is based on its ability to represent the myriad of pathophysiological alterations (not only immune dysregulation) occurring during the evolution of the COVID-19 what might be the base for its good predictive capacity in a very diverse population. Further, CODOP is based on quantitative parameters that are very easy to obtain during the first examination of the patient. Of note, we analyzed the role of twelve patient comorbidities (*Supplementary file 1*) in the performance of CODOP. Interestingly, while six comorbidities were chosen together with the same 12 biochemical parameters, their addition to CODOP did not improve its discriminative ability (*Supplementary file 1*). The impossibility to evaluate this model with the six comorbidities in the external test and evaluation cohorts (due to the lack of these variables) made us decide not to include them in the final online predictor.

The quality, availability, and consistency of biomedical data make reproducibility very challenging for machine learning tools applied to health (*McDermott et al., 2021*) (MLH). The reproducibility of MLH is of critical importance as predictions can affect human health care. Careful analysis indicates that CODOP fulfils the main performance criteria reached in other machine learning subfields when analysing the three main reproducibility principles. In comparison to previous studies, CODOP excels in the 'Conceptual Reproducibility or Replicability' due to the use of geographically spread cohorts (*McDermott et al., 2021*).

The overall performance of CODOP has inherent limitations, some of them generalizable to any MLH. On the one side, the use of training and test datasets with a high degree of perturbations (see above) adds several sources of variability (*Aarsand et al., 2019*): pre-analytical due to differences in blood sampling, analytical due to different laboratory protocols, intra- and inter-individual, and inter-hospital and geographical differences in clinical practices. As an additional factor, the high diversity of COVID-19 encompassing more than 60 disease subtypes (*DeMerle et al., 2021*) sets a limitation in terms of the discriminability ability and the overall clinical utility of any MHL. In contrast to other predictors, CODOP does not take into account the level of care received by each patient (e.g. ICU versus basic care), which influences the outcome of the patient and perturbs the discrimination ability of CODOP (as predictions are made with the data from blood analyses at hospital admission).

The proposed objective of CODOP is not to indicate a specific clinical treatment or decision (e.g. yes/no admission to ICU), but rather to inform physicians about the monitoring needs of patients (i.e. a higher death risk score suggests a closer monitorization of the patient). In this line, the analysis of a small court of patients (belonging to the Test 4 and the Argentinian COVID-19 Network cohorts) shows that CODOP is less effective for predicting the admission of a patient to the ICU or the need for mechanical ventilation (*Supplementary file 1t*). Larger and more geographically diverse datasets are needed to find other parameters that could be the bases to better predict these clinical outcomes. Based on the ability of CODOP to stratify the severity of patients, CODOP could be an interesting tool to increase the number of expected critical events, therefore potentially reducing the sample size calculation. In the same line, CODOP could be useful for the analysis of observational studies.

The clinical utility of MHL has to take into account the changing pressure supported by hospitals during the successive pandemic waves. Our data support the strategy of using either CODOP-Unt or CODOP-Ovt as an effective first-line triage tool in the overall clinical decision procedure. We expect that future participation of more institutions from regions non-represented in our study (Africa, Asia) will improve the reproducibility and overall clinical utility of CODOP supporting subgroup-specific predictions (e.g. based on underlying comorbidities or ethnical background).

## Data sharing

The raw patient data used in this study are not freely available due to legal restrictions of the ethical committees of the different hospitals. However, they can be accessed upon request to the Scientific Committees of these organisms. An exception to this is the patient data from the USA cohort, which has been published elsewhere (*Del Valle et al., 2020*).

All the model's numerical data necessary to generate all figures can be found in the submitted source data tables. Furthermore, all supplementary tables can be found in *Supplementary file 1*.

## Acknowledgements

We gratefully acknowledge all the investigators and staff from the SEMI-COVID-19 Registry, from the Argentinian COVID-19 Network, From the 12 de Octubre and Costa del Sol hospitals, and from the five Latin American hospitals who participate in the collection of the patient data (see Appendix 1).

We also gratefully thank the Data Protection Office of the Max Planck Society which assisted in assuring the legal fit of the web application to the General Data Protection Regulation (GDPR), and to the IT team of the Max Planck Institute of Experimental Medicine for their support in setting up and maintaining the CODOP web calculator.

## Additional information

### Funding

| Funder | Grant reference number | Author |
|---|---|---|
| Max Planck Society | Publication cost | David Gómez-Varela |

The funders had no role in study design, data collection and interpretation, or the decision to submit the work for publication.

### Author contributions

Riku Klén, Data curation, Formal analysis, Investigation, Methodology, Validation, Visualization, Writing – review and editing; Disha Purohit, Data curation, Formal analysis; Ricardo Gómez-Huelgas, Juan Miguel Antón-Santos, Jesús Millán Núñez-Cortés, Carlos Lumbreras, José Manuel Ramos-Rincón, Pablo Young, Resources; José Manuel Casas-Rojo, Noelia García Barrio, Miguel Pedrera-Jiménez, Antonio Lalueza Blanco, María Dolores Martin-Escalante, Francisco Rivas-Ruiz, Maria Ángeles Onieva-García, Juan Ignacio Ramirez, Estela Edith Titto Omonte, Rosmery Gross Artega, Magdy Teresa Canales Beltrán, Pascual Ruben Valdez, Florencia Pugliese, Rosa Castagna, Ivan A Huespe, Bruno Boietti, Javier A Pollan, Data curation, Resources; Nico Funke, Benjamin Leiding, Software; David Gómez-Varela, Conceptualization, Data curation, Formal analysis, Funding acquisition, Investigation, Methodology, Project administration, Resources, Software, Supervision, Validation, Visualization, Writing – original draft, Writing – review and editing

### Author ORCIDs

Disha Purohit http://orcid.org/0000-0002-1442-335X
Juan Miguel Antón-Santos http://orcid.org/0000-0003-3443-1100
José Manuel Ramos-Rincón http://orcid.org/0000-0002-6501-9867
David Gómez-Varela http://orcid.org/0000-0003-2502-9419

### Decision letter and Author response

Decision letter https://doi.org/10.7554/eLife.75985.sa1
Author response https://doi.org/10.7554/eLife.75985.sa2

## Additional files

### Supplementary files
- Transparent reporting form
- Supplementary file 1. Supplementary Tables.
- Supplementary file 2. List of Collaborators.

### Data availability
The raw patient data used in this study are not freely available due to legal restrictions of the ethical committees of the different hospitals. However, they can be accessed upon request to the Scientific Committees of these organisms. An exception to this is the patient data from the USA cohort, which has been published elsewhere. All the model's numerical data necessary to generate all figures can

be found in the submitted source data tables. Furthermore, all supplementary tables can be found in Supplementary File 1.

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
