## [Editor Report]

This submission is dealing with the unmet need to generate a machine learning approach for the early and accurate estimation of the outcome of patients admitted to hospital with COVID-19. The presented data generate confidence on the validity since they have been developed based on a vast number of patients and the data are validated in cohorts from different geographical regions.

---

## [Decision Letter]

**Decision letter after peer review:**

Thank you for submitting your article "Development and evaluation of a machine learning-based in-hospital COVID-19 Disease Outcome Predictor (CODOP): a multicontinental retrospective study" for consideration by *eLife*. Your article has been reviewed by 3 peer reviewers, including Evangelos J Giamarellos-Bourboulis as Reviewing Editor and Reviewer #1, and the evaluation has been overseen by Jos Van der Meer as the Senior Editor. The following individual involved in review of your submission has agreed to reveal their identity: Jesus Bermejo-Martin (Reviewer #3).

Essential revisions:

– The figures in the core of the manuscript are much influenced from the IT output and may generate ambiguity to the non-expert clinician to which the study is addressed. Figure 3A does not offer much. Figure 3B should make clear that high-risk and low-risk refers to the probability provide by the machine output.

– Supplementary Table 1 should move to the core of the text.

– The authors need to explain how the algorithm is applied in real life and the time to result. Information is also needed on how user-friendly this approach is.

– Can the authors provide some analysis if this risk identification approach surrogates for some specific treatment?

– The authors chose only age and laboratory parameters; they should explain why no comorbidities were considered for the model. Second, it would be interesting to provide information about how the model performs with other adverse outcomes except for mortality, such as intubation and ICU admission.

– The authors should discuss more about advantages/disadvantages of the model compared to single biomarkers of prediction (CRP, Il-6, sUPAR), some of which have been already used not only to predict but also to guide COVID-19 treatment (like sUPAR).

– The authors developed two sub-models, CODOP-OVT and CODOP-Unt to be applied in two different contexts depending on the degree of pressure on the hospitals attending COVID-19 patients. In this particular regard, the performance of CODOP-Unt regarding sensibility is far to be acceptable for real applications in clinical practice. In consequence, I do not understand what CODOP-Unt could be useful for, in reality. Can the authors comment on this point?

– Although the authors propose CODOP mostly to predict mortality and decide resources allocation, given the good performance of CODOP to stratify severity, it could also be useful in the design of clinical trials to include patients with similar severity in both the placebo or test groups. CODOP could serve thus as a kind of predictive enrichment tool in this scenario. Please discuss.

– The work was developed in the pre-omicron era. Although multi-level, multinational validation of CODOP seems to ensure their potential application in this scenario, do the authors think that its performance could be affected by Omicron?

– How the progressive extension of vaccination could affect the performance of CODOP, since vaccination could alter disease severity at clinical presentation?

– I would suggest a sub-analysis depending on the vaccinations status, if available, or at least a comment considering the percentage of vaccination in the involved countries in the time when CODOP was derived.

– It would be interesting to evaluate the performance of CODOP separately in men and women.

---

## [Author Response]

Essential revisions:– The figures in the core of the manuscript are much influenced from the IT output and may generate ambiguity to the non-expert clinician to which the study is addressed. Figure 3A does not offer much. Figure 3B should make clear that high-risk and low-risk refers to the probability provide by the machine output.

We thank the reviewer for the suggestions.

We respectfully believe that Figure 3A is important for informing the clinicians about the time window, before clinical resolution, at which CODOP can predict the death of hospitalized patients with high sensitivity. We have now specified this goal in the new revised text (highlighted in yellow; page 8).

We have incorporated the suggestion of the reviewer in the legends of Figure 3B, Figure 3—figure supplement 1, as well as in the main text (highlighted in yellow on page 8 of the revised version of the manuscript).

– Supplementary Table 1 should move to the core of the text.

We have moved Supplementary Table 1 to the main text an named it as Table 1.

– The authors need to explain how the algorithm is applied in real life and the time to result. Information is also needed on how user-friendly this approach is.

Our clinical collaborators from SEMI and FIMI (Latin American hospitals) have guided the design and the validation of this web calculator (https://gomezvarelalab.em.mpg.de/codop/) by taking into account the real-life constraints of their daily clinical practice. As indicated in the manuscript (page 8), the web application includes a detailed description of the CODOP project and instructions on how to use the prediction tool.

In short, CODOP offers an easy-to-use framework with the following steps:

a) The user has the opportunity to choose between CODOP-Ovt and CODOP-Unt in order to adjust for the clinical context (i.e., availability of resources).

b) The user introduces the values of the 12 variables either manually in the provided table (this is an option offered if the user would like to predict the death risk of a few patients), or by uploading a previously-prepared data table (this is an option offered for users that would like to predict the death risk of many patients). As a result, a binary prediction (high or low risk of death) is generated.

c) Following, a table with the original uploaded data plus the new predictions is created and can be downloaded in several formats.

Our performance tests indicate that predictions are obtained in less than 2 seconds (Page 8 of the revised version).

Finally, the web calculator does not store any patient information (therefore, fulfilling the GDPR) and has been tested with several Internet navigators on different operative Systems (Supplementary file 1l of the revised manuscript).

– Can the authors provide some analysis if this risk identification approach surrogates for some specific treatment?

The predictive score provided by CODOP is useful for a detailed and continue monitoring of the disease due to its good predictive performance and its dependency on simple data that is easy to obtain when the patient is admitted. Thus, the proposed objective of CODOP is not to indicate a specific treatment, but rather to inform physicians about the monitoring needs of patients (i.e., a higher death risk score suggests a closer monitorization of the patient).

We now comment on this point in the Discussion section (highlighted in yellow on page 10).

– The authors chose only age and laboratory parameters; they should explain why no comorbidities were considered for the model.

The main reasons for solely considering laboratory parameters were the following:

1) The information regarding comorbidities was not available for all datasets included in this study. While this information was available for the SEMI-COVID dataset, it was not present in the external datasets (online validation datasets from Latin America and New York) that were key for the validation of CODOP.

2) Although not mentioned in the manuscript, we used the SEMI-COVID dataset for testing the influence of several comorbidities together with the whole panel of laboratory parameters indicated in the study. The following comorbidities were considered during the model building process: Diabetes, Cardiomyopathy, EPOC, Dementia, Hypertension, Stroke, Asthma, Cancer, Hyperlipidemia, Chronic Kidney Disease, Chronic Renal Disease, and Sleep Apnea. From these, only Cardiomyopathy, Stroke, EPOC, Dementia, Cancer, and Chronic Kidney Disease were selected after 100 iterations (see Model building procedure in Material and Methods). However, the performance of CODOP is similar when comparing a CODOP model with or without these 6 comorbidities (AUROC 0.893 vs. 0.889, respectively).

We now present these data in the revised version of the manuscript (Supplementary file 1r and 1s) and discuss the potential reasons and implications in the Discussion section (highlighted in yellow on pages 9-10).

Second, it would be interesting to provide information about how the model performs with other adverse outcomes except for mortality, such as intubation and ICU admission.

We thank the reviewer for the suggestion.

Neither ICU admission nor intubation data were present in the SEMI-COVID dataset that was made available to us.

To answer the reviewer’s request, we have incorporated three new datasets from the 12 de Octubre Hospital, the Costa del Sol Hospital, and the Argentinian COVID-19 Network database – all of them having information regarding admission to the ICU and the need of mechanical ventilation (MV). In total, these new datasets added 7198 new patients to our study, which were hospitalized from 2021-22 in 39 health centers.

For the sake of simplicity, we have combined the two Spanish datasets and named Test 4.

CODOP continues to have good performance for predicting in-hospital death, similarly to the datasets presented in the previous version of the manuscript (AUROC: 0.81 for Test 4 and 0.79 for the Argentinian COVID-19 Network database). However, CODOP it is less effective for predicting ICU admission (AUROC: 0.53 for Test 4 and 0.70 for the Argentinian COVID-19 Network database) and the need for MV (AUROC: 0.60 for Test 4 and 0.65 for the Argentinian COVID-19 Network database).

We now present these new data in the revised version of the manuscript (Supplementary file 1t) and commented in the Discussion section (highlighted in yellow on page 10).

– The authors should discuss more about advantages/disadvantages of the model compared to single biomarkers of prediction (CRP, Il-6, sUPAR), some of which have been already used not only to predict but also to guide COVID-19 treatment (like sUPAR).

Several unique biomarkers have been suggested as surrogates for guiding clinical decisions in COVID-19 patients. Generally, the main drawbacks of these studies are the lack of validation in different populations/health systems and/or the limited clinical utility due to other factors, particularly in low-income countries (i.e., elevated price of the test). The advantage of CODOP is that it has a good predictive capacity in a very diverse population and it is based on parameters that are very easy to obtain during the first examination of the patient (at least in all the hospitals from the countries that validated CODOP).

As the reviewer pointed out, C-reactive Protein (CRP), Interleukin 6 (IL-6), or suPAR are useful to recognize patients with increased risk of mortality during hospitalization (Stringer et al., Coomes et al., Altintas et al.). CRP is one of the variables incorporated in CODOP. However, the 12 parameter CODOP model has higher predictive performance than the CRP when analyzed as univariate (see Supplementary Table 3 of the revised manuscript). In our opinion, the better performance of such a multi-parameter predictor is based on the ability to better represent the myriad of pathophysiological alterations (not only immune dysregulation) occurring during the evolution of the COVID-19.

The IL-6 and suPAR are biomarkers associated with adverse outcomes, too. Our clinical collaborators confirmed that none of them is normally used in their Emergency Departments mainly due to their elevated price.

Following the suggestion from the reviewer, we have included a new paragraph in the Discussion section (highlighted in yellow on page 9).

– The authors developed two sub-models, CODOP-OVT and CODOP-Unt to be applied in two different contexts depending on the degree of pressure on the hospitals attending COVID-19 patients. In this particular regard, the performance of CODOP-Unt regarding sensibility is far to be acceptable for real applications in clinical practice. In consequence, I do not understand what CODOP-Unt could be useful for, in reality. Can the authors comment on this point?

During this pandemic, it has not been always possible to provide appropriate resources/treatment for all patients – a situation reached in limited resource places, particularly. In this scenario, undertriage was necessary. As a consequence, some patients at high risk of death will not receive adequate treatment. A predictive tool that could maximize the utilization of scarce resources will have a clear benefit.

For this reason, and as a request from some of our collaborators in Latin America, we set up CODOP-Unt for optimizing the identification of patients with the lowest risk of death (therefore, not being ranked as high priority patients for the use of limited resources). CODOP-Unt maximizes the positive predictive value by trying to avoid the inclusion of false high-risk patients (high specificity) and it might be preferred in pandemic conditions when hospital resources are limited and undertriage needs to be considered. As a result of maximizing the specificity, the sensitivity is significantly reduced.

We believe that CODOP-Unt could be considered as an additional tool in the final triage decisions together with other standardized parameters included in the current triage protocols of each hospital.

– Although the authors propose CODOP mostly to predict mortality and decide resources allocation, given the good performance of CODOP to stratify severity, it could also be useful in the design of clinical trials to include patients with similar severity in both the placebo or test groups. CODOP could serve thus as a kind of predictive enrichment tool in this scenario. Please discuss.

We thank the reviewer for the suggestion.

We totally agree with the reviewer – CODOP could be an interesting alternative to randomization in the design of clinical trials for including patients with similar severity in both the placebo and test groups. Additionally, using CODOP to include patients with higher severity in a clinical trial will increase the number of expected critical events, therefore potentially reducing the sample size calculation. Another interesting possibility is the application of CODOP in observational studies.

We now discuss these possibilities in the Discussion section (highlighted in yellow on page 10).

– The work was developed in the pre-omicron era. Although multi-level, multinational validation of CODOP seems to ensure their potential application in this scenario, do the authors think that its performance could be affected by Omicron?

We thank the reviewer for pointing out this important point, which has prompted us to look for new data on patients infected with Omicron.

The newly incorporated dataset Test 4 (please, see above) encompass the whole Δ wave and more than two months of the Omicron (BA.1) wave in Spain (from Dec 15^th^, 2022 until Feb 27th, 2022). Our analysis (a total of 1728 patients for Δ and 1010 patients for Omicron) shows that CODOP is equally predictive as for the previous Variants Of Concern (AUROC: Δ 0.79, Omicron 0.82).

We now present these new data in the revised version of the manuscript (highlighted in yellow in page 7, new Figure 2—figure supplement 2, Figure 2—figure supplement 3, Supplementary file 1d and Supplementary file 1e), as well as discussing the potential reasons and implications (highlighted in yellow on page 9).

– How the progressive extension of vaccination could affect the performance of CODOP, since vaccination could alter disease severity at clinical presentation?– I would suggest a sub-analysis depending on the vaccinations status, if available, or at least a comment considering the percentage of vaccination in the involved countries in the time when CODOP was derived.

We thank the reviewer for the suggestion.

The information about the vaccination status was not was present in any of the datasets that were available to us.

To evaluate the predictive performance of CODOP in the vaccinated population we analyzed the newly incorporated datasets Test 4 (accounting for a total of 310 and 125 vaccinated patients with at least two or three doses of a COVID-19 vaccine, administered before hospital admission). Our data show that CODOP is similarly effective in predicting in-hospital mortality independently of the vaccination status of the patient.

These results suggest that, despite the reported benefit of vaccination in decreasing COVID-19 mortality at the population level, the inflammatory process underlying the clinical manifestations in a percentage of vaccinated patients are comparable to non-vaccinated patients.

We now present these new data in the revised version of the manuscript (new Figure 2—figure supplement 2 and Supplementary file 1d; Of note: the step-wise appearance of the ROC curves in Figure 2—figure supplement 2 is typical when a low number of patients is included in a dataset) and comment on the potential reasons and implications in the Discussion section (highlighted in yellow on page 9).

– It would be interesting to evaluate the performance of CODOP separately in men and women.

We thank the reviewer for the suggestion.

A new analysis demonstrates a similar performance of CODOP in men (N=9127, AUROC=0.883 (0.877-0.888)) and women (N=6775, AUROC=0.893 (0.888-0.905)), which goes in line with the not selection of Sex as predictive parameter during our model building process (Supplementary file 1a).